# InteractComp: Evaluating Search Agents With Ambiguous Queries

Mingyi Deng [* 1]  Lijun Huang [* 2]  Yani Fan [2]  Fanqi Kong [3 1]  Jiayi Zhang [2]  Fashen Ren [2]  Jinyi Bai [4]
Fuzhen Yang [4]  Dayi Miao [2]  Zhaoyang Yu [1]  Yifan Wu [2]  Yanfei Zhang [1]  Fengwei Teng [1]  Yingjia Wan [1 5]
Song Hu [1]  Yude Li [1]  Xin Jin [1]  Conghao Hu [1]  Haoyu Li [1]  Qirui Fu [1]  Tai Zhong [6]  Xinyu Wang [7]
Xiangru Tang [8]  Nan Tang [2]  Chenglin Wu [1]  Yuyu Luo [2]

## Abstract

Language agents have demonstrated remarkable potential in web search and information retrieval. However, many search-agent benchmarks assume that user queries are complete and unambiguous. This assumption leaves under-tested a practical failure mode: agents may face ambiguous requests where the intended target cannot be identified without clarification. Yet most agents lack interactive mechanisms during the search process, and existing benchmarks cannot assess this capability. To address this gap, we introduce INTERACTCOMP, a benchmark designed to evaluate whether search agents can recognize query ambiguity and actively interact to resolve it during search. Following the principle of **easy to verify, interact to disambiguate**, we construct 210 expert-curated questions across 9 domains through a target-distractor methodology that creates controlled ambiguity resolvable only through interaction. Evaluation of 17 models reveals striking failure: the best model achieves only 13.73% accuracy despite 71.50% with complete context, exposing systematic overconfidence rather than reasoning deficits. Forced interaction produces dramatic gains, demonstrating latent capability current strategies fail to engage. Longitudinal analysis shows interaction capabilities stagnated over 15 months while search performance improved seven-fold, revealing a critical blind spot. This stagnation, coupled with the immediate feedback inherent to search tasks, makes INTERACTCOMP a valuable resource

for both evaluating and training interaction capabilities in search agents. The code is available at https://github.com/FoundationAgents/InteractComp.

## 1. Introduction

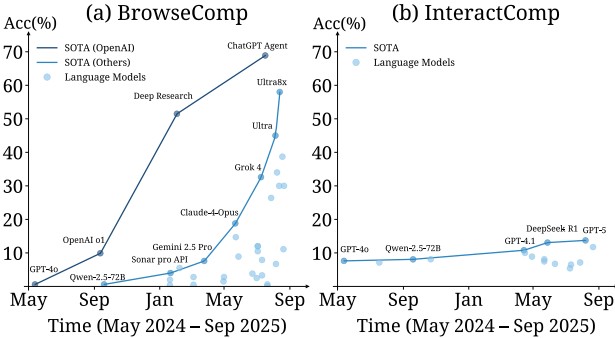

*Figure 1.* Despite rapid progress on complete search queries (BrowseComp: seven-fold over 15 months), agent performance on ambiguous, interaction-dependent queries (INTERACTCOMP) has stagnated around 6-14%. This growing disparity reveals a critical blind spot in agent development.

Language agents have demonstrated remarkable potential across diverse domains, including code generation (Zhang et al., 2025b; Hong et al., 2024b) , data analysis (Hong et al., 2024a; Li et al., 2025b;a), information retrieval (Geng et al., 2025; Song et al., 2025), and decision-making (Liu et al., 2025a; Liang et al., 2025). A notable trend is the rapid development of search agents (OpenAI, 2025d; Google, 2025b), which can handle complex user queries and gather information across the internet by performing search, browse, and reasoning actions (Mialon et al., 2023; Wei et al., 2025).

However, these advanced search agents assume user queries are complete and unambiguous. In practice, users begin with incomplete queries admitting multiple plausible interpretations, and only through interaction can the true intent be identified. Yet most search agents lack interactive mechanisms during search. Commercial agents (OpenAI, 2025d) engage in a single clarification, with no further interaction

---

[*]Equal contribution  [1]DeepWisdom  [2]The Hong Kong University of Science and Technology (Guangzhou)  [3]Peking University  [4]Renmin University of China  [5]University of California, Los Angeles  [6]AgentUniverse  [7]McGill University  [8]Yale University. Correspondence to: Jiayi Zhang <jzhang361@connect.hkust-gz.edu.cn>.

*Proceedings of the 43rd International Conference on Machine Learning*, Seoul, South Korea. PMLR 306, 2026. Copyright 2026 by the author(s).

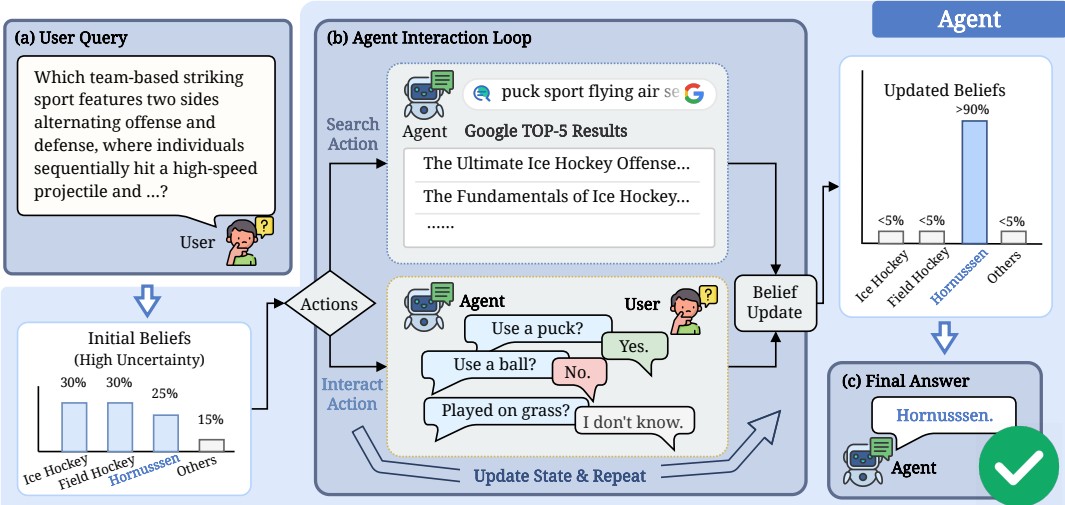

*Figure 2.* An example interaction in INTERACTCOMP. Given an ambiguous user query, the agent iteratively interacts with the user through clarification questions and external search actions, and produces a final answer after resolving key uncertainties.

once search begins. When faced with ambiguity, agents confidently commit to assumed queries, leading to incorrect answers and wasted computational resources.

Existing benchmarks cannot assess this capability. Search benchmarks like GAIA (Mialon et al., 2023) and BrowseComp (Wei et al., 2025) provide all necessary resources upfront, enabling agents to proceed without clarifying ambiguous intent. Interaction benchmarks like IN3 (Qian et al., 2024) and Tau-Bench (Yao et al., 2024) focus on general conversation but lack grounding in verifiable search tasks. Neither addresses the question: *Can agents recognize query ambiguity and actively interact to gather disambiguating information during search?* Without proper assessment of this capability, we cannot determine whether recent advances in search agents translate to handling real-world scenarios where user intent must be uncovered rather than assumed.

Motivated by this gap, we introduce INTERACTCOMP, a benchmark designed to evaluate whether search agents can recognize ambiguity and actively interact to resolve it. As illustrated in Figure 2, our benchmark workflow demonstrates how agents navigate from ambiguous queries to definitive answers through structured interaction. Our design follows a core principle: **easy to verify, interact to disambiguate**. Questions have short, verifiable answers (1-2 words) that are answerable with enough context, yet require interaction to obtain specific details needed for disambiguation. Through the interaction loop, agents must iteratively update their initial beliefs, which typically exhibit high uncertainty across multiple candidate answers, by gathering clarifying information from users, progressively refining their confidence until a clear answer emerges. We achieve this through a

target-distractor design: questions use only shared attributes of a lesser-known target and a popular alternative, creating genuine ambiguity that search alone cannot resolve. Agents must interact with simulated users to uncover distinctive attributes not given in the initial query. Referring to some influential benchmarks, such as HumanEval (Chen et al., 2021)(164 hand-crafted programming challenges;), BrowseComp-ZH (Zhou et al., 2025a)(289 multi-hop questions), INTERACTCOMP contains 210 expert-curated questions across 9 domains in both English and Chinese, validated to ensure interaction is necessary and answers are verifiable.

Systematic evaluation of 17 models confirms our design principle and reveals a striking failure pattern. When provided complete disambiguating context, models achieve strong performance with the best reaching 71.50% accuracy, validating questions are answerable once information is complete. However, even the best model achieves only 13.73% in the full interaction setting, with most models in single digits. This 5× performance gap exposes the core problem: models fail not due to search and reasoning deficits, but systematic overconfidence that prevents them from engaging in interaction despite having access to it.

Scaling experiments confirm this diagnosis. Simply increasing interaction opportunities from 5 to 20 rounds yields minimal improvement (from 14% to 20%), as models barely increase their interactive behavior. In contrast, forcing models to interact before answering produces dramatic gains (from 14% to 40%). Longitudinally, as shown in Figure 1, interaction capabilities have shown almost no improvement across all models over 15 months, while BrowseComp per-

formance improved seven-fold during the same period. This stagnation is striking given our forced interaction experiments demonstrate the capability is latent rather than absent and readily improvable. This finding, combined with the clean reward signals from search outcomes, makes INTER-ACTCOMP well-suited for recent environment-driven and verifiable agent-training paradigms (Zhang et al., 2025a; 2026b; Wu et al., 2026), including Reinforcement Learning from Verifiable Rewards (RLVR), where models are trained using binary correctness signals from verifiable outcomes. More broadly, it also provides a controlled setting for studying how agent capabilities can be improved through automated or evolutionary training procedures (Zhang et al., 2026a).

Our contributions are threefold. (1) We introduce INTER-ACTCOMP, a benchmark evaluating interaction capabilities in search scenarios, with clean reward signals enabling future training approaches. (2) We provide diagnostic evidence across 17 models that interaction failure stems from systematic overconfidence rather than capability deficits. (3) We demonstrate through longitudinal analysis that interaction represents a critical blind spot in agent development, with INTERACTCOMP providing a foundation for addressing this neglected dimension.

## 2. Related Work

**Search Benchmarks and Agents.** Recent benchmarks evaluate search agents along two dimensions. Web-scale search benchmarks like BrowseComp (Wei et al., 2025) assess information gathering across the entire web with complete queries, spawning variants for Chinese (Zhou et al., 2025a), multimodal content (Li et al., 2025d), and enhanced questions (Chen et al., 2025). Tool-augmented benchmarks like GAIA (Mialon et al., 2023) and WebWatcher (Geng et al., 2025) additionally require agents to handle multimedia and perform computations. Beyond web and tool-use settings, evaluation work has also studied multimodal analytical reasoning, such as low-level chart understanding (Wu et al., 2024). These benchmarks have motivated diverse agent designs. Reinforcement learning approaches like R1-Searcher (Song et al., 2025) and Search-R1 (Jin et al., 2025) learn integrated search-reasoning patterns, while data synthesis methods like WebSailor (Li et al., 2025c) and WebExplorer (Liu et al., 2025b) enhance long-horizon capabilities. Additionally, both manually designed and self-designed search agents (Zhang et al., 2025b; Zeng et al., 2025; Teng et al., 2025) have achieved strong performance through careful workflow engineering. Datasets such as Search Arena (Miroyan et al., 2025), and WildChat (Zhao et al., 2024) further show that real user queries are often under-specified, motivating benchmarks that explicitly evaluate ambiguity resolution.

**Interaction Benchmarks and Agents.** Prior work has extensively studied ambiguity and clarification in open-domain question answering and information seeking. AmbigQA (Min et al., 2020) constructs disambiguated question-answer pairs for ambiguous open-domain questions; Yu et al. (2020) formulates question asking as uncertainty reduction; Aliannejadi et al. (2019) studies clarifying questions in open-domain information-seeking conversations; and Stelmakh et al. (2022) further investigates ambiguity in open-domain QA. Complementary to search benchmarks, recent work evaluates agents' interaction capabilities in more dynamic settings. SWEET-RL (Zhou et al., 2025b) proposes ColBench for multi-turn collaborative reasoning with RL-based credit assignment across turns. UserBench (Qian et al., 2025a) and UserRL (Qian et al., 2025b) create gym environments for training agents on user-centric tasks where goals are underspecified and preferences emerge incrementally. IN3 (Qian et al., 2024) and Tau-Bench (Yao et al., 2024) evaluate implicit intention understanding and tool-agent-user interaction respectively. These benchmarks collectively reveal that current models struggle with proactive clarification and user alignment—for instance, agents uncover fewer than 30% of user preferences through active questioning in UserBench.

However, INTERACTCOMP is complementary to these works: rather than treating ambiguity resolution as a standalone QA or clarification task, it embeds ambiguity into a search-agent loop where the model must choose among searching, asking, and answering, with final answers remaining short and verifiable. It differs by evaluating interaction capabilities specifically in search scenarios, where ambiguous queries must be resolved through clarification before effective retrieval can occur, and where search outcomes provide natural reward signals for training interaction strategies.

## 3. The INTERACTCOMP Benchmark

The INTERACTCOMP dataset was constructed entirely by human annotators with the assistance of search tools and language models. While BrowseComp (Wei et al., 2025) evaluates complex search and reasoning with complete initial information, INTERACTCOMP evaluates whether agents can recognize ambiguity and actively gather necessary context through interaction during the search process. Our core design principle follows "**Easy to verify, Interact to disambiguate**": questions have concise answers that are straightforward to verify once found, yet remain ambiguous without interaction to uncover distinguishing details. This section describes the task structure (§3.1), construction methodology (§3.2), and dataset statistics (§3.3).

**Algorithm 1** Data Construction Pipeline

---

1: **Input:** target $A$, distractor $B$
2: $F_A \leftarrow$ attributes of $A$;  $F_B \leftarrow$ attributes of $B$
3: Build ambiguous $Q$ from $F_A \cap F_B$
4: Add context $C$ from $F_A \setminus Q$
5: Validate $(Q, C)$
6: **while** not finished **do**
7: **if** candidates $\geq 4$ **or** $Q$ answerable **then**
8:  refine $Q$
9: **else**
10:  **if** answer not unique **then**
11:   refine $C$
12:  **else**
13:   **if** cross-validation fails **then**
14:    repair $Q$ **or** $C$
15:   **end if**
16:  **end if**
17: **end if**
18: **end while**
19: **Output:** finalized instance $(Q, C, A)$

---

### 3.1. Task Overview

As shown in Appendix A.1, each instance comprises an ambiguous question, a context containing distinctive attributes, the correct answer, and a distractor (a popular alternative sharing attributes with the target). The context is hidden from agents but available to a simulated user responder. Agents receive only the ambiguous question and operate with three actions: `search` to retrieve web information, `interact` to propose clarifying questions, and `answer` to provide the final response. We intentionally adopt a binary yes/no interaction protocol not to simplify the task, but to prevent interaction from collapsing into unrestricted information retrieval. This design forces agents to commit to specific hypotheses and tests whether they can identify the right questions to ask, rather than merely asking more questions. The simulated responder replies with "yes", "no" or "I don't know" based solely on context information.

Implementation details for both agents and responders are provided in Appendix A.2 and Appendix A.3.

### 3.2. Data construction and Verification

Our construction methodology draws inspiration from BrowseComp's answer-first approach (Wei et al., 2025), but fundamentally shifts focus from search complexity to ambiguity resolution. The central challenge in constructing such a benchmark is creating questions that appear reasonable yet systematically lack information for confident resolution. As a related line of work examining tip-of-the-tongue retrieval (CH-Wang et al., 2025; Arguello et al., 2021), where users provide incomplete or noisy descriptions due to memory failure We observe that user ambiguity is particularly pronounced when dealing with similar concepts that share overlapping attributes, it is in these scenarios that additional clarification becomes truly necessary rather than merely helpful.

This observation leads us to design a systematic target-distractor methodology. We deliberately pair a target entity with a similar popular entity (the distractor), crafting questions using only their shared attributes while hiding distinctive information as context. This construction ensures that: (1) questions admit multiple plausible interpretations including the popular distractor, making direct answering unreliable; (2) the target answer possesses all described attributes, ensuring verifiability; and (3) distinctive attributes hidden in context provide clear disambiguation paths through interaction. More generally, our target-distractor construction follows the broader trend of evaluating model reasoning under structured constraints, where success depends on satisfying hidden or constrained conditions rather than surface-level pattern matching (Yang et al., 2026). Algorithm 1 formalizes this pipeline, which we detail in the following subsections alongside our two-stage verification process.

All annotators are master's and Ph.D. students from various universities.

#### 3.2.1. CONSTRUCTION PROCESS

Annotators receive the following instruction:

*"You need to find a pair of entities that are similar but differ in popularity. Use their shared attributes to construct an ambiguous question, and reserve the remaining distinctive attributes to form the context."*

Following this instruction, the construction proceeds in four steps: **(1) Entity Selection**: annotators identify a lesser-known target and a popular distractor sharing overlapping characteristics; **(2) Attribute Categorization**: attributes are classified as shared (common to both) or distinctive (unique to target); **(3) Question Formulation**: only shared attributes are used to create questions admitting multiple plausible candidates; **(4) Context Formation**: distinctive attributes are reserved as context, ensuring question-context pairs uniquely identify the target while questions alone remain ambiguous.

#### 3.2.2. VERIFICATION PROCESS

We implement a two-stage verification protocol to ensure data quality and interaction necessity.

**Stage 1: Completeness Verification.** Independent annotators validate three requirements: (1) the target answer must possess all attributes described in both the question and context, (2) the question-context combination must admit only

one valid answer with no plausible alternatives, and (3) instances where annotators identify valid alternative answers are discarded and reconstructed.

We also let at least 2 annotators cross-paraphrased each question and checking whether the candidate set remains unchanged under different surface forms to guarantee that ambiguity is only caused by *missing essential information*, not by linguistic phrasing.

**Stage 2: Interaction Necessity Validation.** We verify whether questions truly require interaction through two complementary checks. First, we manually confirm questions cannot be confidently resolved through direct web search, checking the first five Google result pages by hand. Second, to ensure genuine ambiguity, we verify that questions cannot be confidently answered without interaction. We conduct automated testing with three capable models (GPT-5, GPT-5-mini, Claude-Sonnet-4) across 5-round trials. Questions successfully answered by two or more models without access to interaction undergo revision to strengthen their ambiguity. This validation tests only whether questions lack discriminative information in isolation, not whether models can gather information through interaction. This stage is exactly where LLMs were only used during data construction.

### 3.3. Data Statistics

**Topic distribution.** Figure 3 presents the distribution of samples across 9 topic domains in the INTERACTCOMP dataset. The most represented categories include Science & Engineering (21.3%), Humanities (18.0%), and Entertainment (16.6%). The dataset also features Business & Economics (11.8%), Law & Politics (8.5%), and Sports (7.1%). Conversely, domains like Medicine & Life Science (5.7%), Academic & Research (4.7%), and General Knowledge & Misc. (6.2%) have fewer samples.

**Question and Context Length distribution.** Figure 3 illustrates the distribution of question and context lengths in the INTERACTCOMP dataset. Question length predominantly ranges between 40 to 80 words, with the majority falling within this interval. Context length shows a broader distribution, typically spanning from 40 to over 200 words, with peak frequency in the 60-100 word range. These distributions demonstrate that questions are concise yet informative, while contexts provide comprehensive disambiguation information.

**Language distribution.** The INTERACTCOMP dataset comprises bilingual instances with English accounting for 139 samples (66.19%) and Chinese contributing 71 samples (33.81%), enabling evaluation of interaction capabilities across different linguistic contexts. In this section, we present statistics on the topic distribution, question and

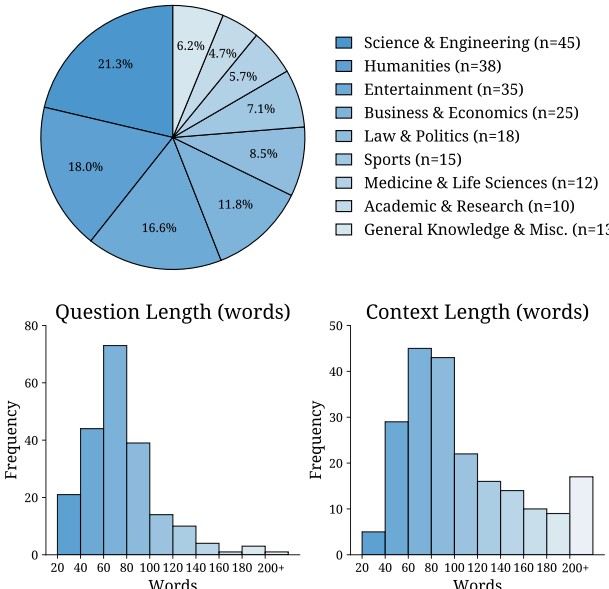

*Figure 3.* Topic distribution and question/context length statistics in INTERACTCOMP.

context length distribution of our curated INTERACTCOMP dataset.

## 4. Experiments

### 4.1. Experimental Setup

To systematically evaluate agent capabilities across different interaction paradigms, we design a controlled experimental framework that isolates and measures the incremental contribution of core agent capabilities: knowledge recall, information retrieval, and interactive clarification.

**Agent Architecture**: We employ the ReAct framework (Yao et al., 2023) as our base architecture, implementing four complementary configurations: (1) *Answer-only*: testing direct response generation; (2) *Answer+Search*: incorporating information retrieval; (3) *Answer+Interact*: allowing interactive clarification without search; and (4) *Answer+Search+Interact*: enabling both retrieval and clarification. This design isolates the incremental contribution of retrieval and interaction while maintaining architectural consistency.

The main results in Table 1 correspond to the Answer+Search+Interact configuration with a maximum of 10 rounds per instance. Additional configuration details are summarized in Appendix A.2.

**Models**: We evaluate across diverse model families including proprietary models (GPT-4o-mini, GPT-4o, GPT-4.1, GPT-5, OpenAI o3, Grok-4, Doubao-1.6, Claude-Sonnet-

4, Claude-Opus-4, Claude-3.5-Sonnet) and open-weight models (GLM-4.5, Kimi-K2, Deepseek-V3.1, Deepseek-R1, Qwen3-235B-A22B, Qwen2.5). Following established benchmarking practices, we standardize parameters where supported: temperature=0.6, top_p=0.95. We implement responder simulation using GPT-4o (temperature=1.0) that provides structured feedback when agents employ the *interact* action. Additional robustness analyses, including responder sensitivity, human validation of the simulated responder, commercial search-agent systems, and ranking consistency, are reported in Appendix B.

**Metrics**: We evaluate agents across five key dimensions: (1) **Interaction Metrics**: Round (average number of conversation turns) and percentage of rounds where interact actions are used (IR) measuring behavioral patterns and action utilization; (2) **Performance Metrics**: Accuracy (Acc.) measuring the percentage of correctly answered queries, and Calibration Error (C.E.) measuring confidence calibration using 5 confidence bins. The precise definition of Calibration Error is provided in Appendix A.6; and (3) **Cost**: measured in USD reflecting computational resources usage for practical deployment considerations.

## 4.2. Main Results

Table 1 corresponds to the Answer+Search+Interact configuration, presenting comprehensive results across 17 models, revealing striking patterns in how different architectures handle ambiguous queries. The results expose fundamental limitations even in state-of-the-art systems, with the highest-performing model (GPT-5) achieving only 13.73% accuracy, demonstrating the benchmark's challenging nature. Because several models differ by only a few accuracy points, we focus on broad performance bands and the large gap to the with-context ceiling rather than fine-grained rankings; Appendix B.4 reports ranking consistency across runs.

**Diverse Interaction Patterns Across Models.** Models exhibit dramatically different interaction strategies, creating distinct behavioral profiles. GPT-4o-mini stands out as an extreme case: it asks questions in 73.95% of available rounds, by far the highest interaction rate, yet achieves only 7.14% accuracy—close to GLM-4.5 which barely interacts (0.25% IR). This suggests that excessive questioning without clear purpose can be counterproductive. Conversely, DeepSeek-R1 demonstrates more balanced behavior with 44.72% IR yielding 13.08% accuracy, the highest among open-weight models, indicating that willingness to interact can translate to better performance when used effectively.

**Calibration Quality Correlates with Interaction Patterns.** A remarkable finding is that models with higher interaction rates often exhibit superior calibration. GPT-4o-mini's aggressive questioning strategy, while not improving accuracy, results in dramatically better calibration (37.44 CE) com-

pared to low-interaction models like Doubao-1.6 (84.35 CE). This pattern suggests that interaction, even when not optimally targeted, helps models develop more realistic confidence assessments about their knowledge limitations.

**Open-Weight vs. Proprietary Model Divide.** The performance gap between open-weight and proprietary models is stark and consistent. All open-weight models struggle with interaction rates below 45%, with most falling under 32%. GLM-4.5, Kimi-K2, and Qwen3-235B-A22B show particularly conservative interaction behavior (0.25%, 5.98%, and 27.75% respectively), suggesting that open-weight models may have been trained to minimize uncertain responses rather than seek clarification. In contrast, proprietary models like GPT-4.1 and GPT-5 show more balanced interaction patterns (34.02% and 30.87%), though even they fall short of optimal information-gathering behavior.

These findings collectively demonstrate that current language models, regardless of scale or sophistication, struggle fundamentally with effective information gathering, often exhibiting either excessive conservatism or ineffective over-questioning when faced with genuine ambiguity.

To systematically understand failure modes, we conduct error attribution analysis on failed instances from three representative models in their *Answer+Search+Interact* trajectories in Section 4.7.

## 4.3. Ablation Analysis

To validate that our benchmark specifically tests interaction abilities rather than general reasoning, we conduct ablation studies across four evaluation modes using 8 representative models.

Table 2 reveals dramatic performance gaps confirming interaction as the critical missing component. Three key findings emerge: (1) *Answer-only* mode exposes fundamental limitations, OpenAI o3 achieves only 5.18%, GPT-5 reaches 7.62%, with catastrophic overconfidence (60.94-93.17% calibration errors). (2)*Answer+Search* provides minimal benefits, o3 increases to just 8.81% and GPT-5 to 9.52%, demonstrating that information retrieval alone cannot resolve ambiguity. (3)*Answer+Interact*(6.67-25.24%) with GPT-5 achieving 25.24% and general improvement in accuracy indicates the importance of interaction when facing the ambiguous questions (4) Complete contextual information reveals the performance ceiling, o3 soars to 71.50% (13.8× increase), GPT-5 reaches 67.88%, and calibration errors plummet to 7.44%, confirming underlying reasoning capabilities exist but are inaccessible without proper context.

Notably, *ask-only*(6.67-25.24%) outperforms *search-only*(6.74-9.52%) not because models can solve the task using internal knowledge, but because they acquire the dis-

*Table 1.* Performance comparison of 17 large language models on the INTERACTCOMP dataset. The table reports both interaction behaviors like average number of conversation turns(Round) and percentage of rounds where interact actions (IR) are used; final performance like accuracy (Acc. with std in parentheses) and calibration error (C.E.), along with the estimated total cost. Models are grouped into *open-weight* and *closed-weight* categories for clarity. Best accuracy is highlighted in bold.

| Model | Interaction | | Performance | | Cost($) |
|---|---|---|---|---|---|
| | Round | IR(%) | Acc.(%) | C.E. | |
| *Open Weights Models* | | | | | |
| GLM-4.5 (Zhipu AI, 2025) | 6.91 | 0.25 | 7.14 (±0.48) | 80.64 | 2.16 |
| Kimi-K2 (Moonshot AI, 2025) | 4.95 | 5.98 | 6.51 (±1.53) | 87.10 | 0.75 |
| Deepseek-V3.1 (DeepSeek, 2025a) | 7.26 | 11.60 | 11.74 (±2.71) | 74.79 | 8.84 |
| Deepseek-R1 (DeepSeek, 2025b) | 6.58 | 44.72 | **13.08** (±0.29) | 77.00 | 60.43 |
| Qwen2.5-72B-Instruct (Yang et al., 2024) | 7.45 | 31.88 | 8.08 (±0.73) | 77.57 | 0.15 |
| Qwen3-235B-A22B (Qwen Team, 2025) | 5.64 | 27.75 | 8.89 (±0.72) | 82.63 | 7.47 |
| *Proprietary Models* | | | | | |
| GPT-4o-mini (OpenAI, 2024b) | 4.16 | 73.95 | 7.13 (±0.42) | 37.44 | 0.35 |
| GPT-4o (OpenAI, 2024a) | 5.65 | 9.26 | 7.62 (±0.79) | 79.50 | 8.65 |
| GPT-4.1 (OpenAI, 2025a) | 5.49 | 34.02 | 10.79 (±1.22) | 82.11 | 5.58 |
| OpenAI o3 (OpenAI, 2025c) | 2.96 | 15.03 | 10.00 (±1.44) | 41.96 | 5.04 |
| GPT-5 (OpenAI, 2025b) | 4.33 | 30.87 | **13.73** (±2.55) | 68.67 | 16.85 |
| Grok-4 (xAI, 2025) | 4.92 | 4.55 | 8.40 (±1.24) | 69.00 | 77.55 |
| Gemini-2.5-Pro (Google, 2025a) | 4.65 | 11.09 | 10.28 (±0.37) | 86.52 | 15.04 |
| Doubao-1.6 (ByteDance, 2025) | 3.08 | 10.60 | 6.73 (±0.97) | 84.35 | 1.40 |
| Claude-3.5-Sonnet (Anthropic, 2024) | 5.63 | 27.57 | 8.10 (±1.91) | 80.04 | 13.09 |
| Claude-Sonnet-4 (Anthropic, 2025b) | 6.90 | 10.76 | 7.46 (±1.37) | 79.62 | 19.47 |
| Claude-Opus-4 (Anthropic, 2025a) | 8.55 | 10.86 | 8.10 (±0.96) | 78.42 | 115.47 |

*Table 2.* Ablation study comparing model performance under four evaluation settings: answer-only (models respond without additional evidence), search-only (responses based solely on retrieved information), ask-only (responses based solely on interaction), and with-context (responses supported by complete disambiguating context). Results are reported in terms of accuracy (Acc.) and calibration error (C.E.). The best scores in each column are highlighted in bold.

| Model | answer-only | | search-only | | ask-only | | with-context | |
|---|---|---|---|---|---|---|---|---|
| | Acc. | C.E. | Acc. | C.E. | Acc. | C.E. | Acc. | C.E. |
| GPT-4o | 2.38 | 88.76 | 7.77 | 80.52 | 7.62 | 72.60 | 40.93 | 47.33 |
| GPT-5 | **7.62** | 76.26 | **9.52** | 79.14 | **25.24** | 57.77 | 67.88 | 21.36 |
| OpenAI o3 | 5.18 | 60.94 | 8.81 | 52.62 | 12.86 | 51.46 | **71.50** | 7.44 |
| GLM-4.5 | 2.38 | 84.40 | 6.74 | 82.41 | 14.29 | 70.98 | 64.77 | 22.37 |
| Kimi-K2 | 1.43 | 90.36 | 7.53 | 86.87 | 6.67 | 86.14 | 53.37 | 40.62 |
| Gemini-2.5-Pro | 2.38 | 93.17 | 7.25 | 90.65 | 12.38 | 84.23 | 69.95 | 28.60 |
| DeepSeek-V3.1 | 3.11 | 85.60 | 8.29 | 79.24 | 15.24 | 67.43 | 65.28 | 24.17 |
| Claude-Sonnet-4 | 2.85 | 87.12 | 7.25 | 81.70 | 15.71 | 65.84 | 59.07 | 26.31 |

ambiguating attributes directly from the context through clarification. Because these attributes are not contained in the model's parametric knowledge and are intentionally absent from the initial query. Therefore, *ask-only* success does not indicate that the task can be solved without search, but that search cannot succeed before disambiguation occurs.

The massive gap between *search-only* (6.74-9.52%) and *with-context* (40.93-71.50%) performance validates our benchmark design: interaction to acquire disambiguating information is the true bottleneck, not reasoning ability. Models possess the knowledge to answer correctly but fail

at recognizing when and how to seek necessary clarification.

### 4.4. Scaling Analysis

The ablation studies revealed that models possess the capabilities to handle ambiguous queries when given complete context, but fail to gather necessary information through interaction. We investigate whether providing more interaction opportunities (5, 10, and 20 rounds) encourages information gathering. Figure 4(a) and Table 3 present the results.

Results show that models fail to scale interaction usage with available opportunities. Despite quadrupling round limits from 5 to 20, GPT-5 increases interactions from just 1.14 to 1.90, while Claude-Sonnet-4 barely reaches 0.78 interactions per instance. However, models that do interact more achieve better performance. GPT-5 improves from 14.00% to 20.00% accuracy as interactions increase. This reveals systematic overconfidence as the primary bottleneck: models prematurely conclude they have sufficient information despite evidence that continued questioning improves performance.

*Table 3.* Scaling analysis of model performance across 3 different interaction budgets (5, 10, and 20 rounds) on a 50-question subsample. We report accuracy (Acc.) and the average number of interaction rounds used (IRound) for four representative models: GPT-4o-mini, GPT-5, Claude-Sonnet-4, and Deepseek-V3.1.

| Model | 5 Rounds | | 10 Rounds | | 20 Rounds | |
|---|---|---|---|---|---|---|
| | Acc. | IRound | Acc. | IRound | Acc. | IRound |
| GPT-4o-mini | 4.00 | 2.00 | 8.00 | 3.62 | 8.00 | 2.76 |
| GPT-5 | 14.00 | 1.14 | 16.00 | 1.76 | 20.00 | 1.90 |
| Claude-Sonnet-4 | 6.00 | 0.16 | 4.00 | 0.70 | 8.00 | 0.78 |
| DeepSeek-V3.1 | 10.00 | 0.38 | 8.00 | 0.74 | 10.00 | 1.54 |

*Table 4.* Natural language interaction performance on INTERACT-COMP. We report accuracy (Acc.), calibration error (C.E.), ask ratio (Ask%), and average interaction rounds (IRound) for four representative models.

| Model | Acc. | C.E. | Ask% | IRound |
|---|---|---|---|---|
| GPT-5 | 25.71 | 64.54 | 28.08 | 4.83 |
| Claude-Sonnet-4 | 7.62 | 78.49 | 14.58 | 7.64 |
| DeepSeek-V3.1 | 17.14 | 70.22 | 5.21 | 8.41 |
| Gemini-2.5-Pro | 11.43 | 86.48 | 7.11 | 4.69 |

### 4.5. Natural Language Interaction Analysis

To isolate the protocol restrictions influence, we implement an askNL mode where the responder's output is no longer restricted to {yes, no, I don't know} but instead provides free-form natural language answers. We note that yes/no responses are also natural language; the distinction here is whether the responder is constrained to a closed three-way choice. This serves as a controlled experiment: if information bandwidth is the primary bottleneck, askNL should approach with-context performance; if recognition failure dominates, the gap should persist.

As Table 4 shows, while natural language interaction improves performance (GPT-5: 25.71%, DeepSeek: 17.14%), all models remain far below their *with-context* ceilings (59-72%). GPT-5's 42% gap (25.71% vs 67.88%) is particularly telling: despite access to richer responses, the model still fails to identify discriminative attributes through interaction.

### 4.6. Forced Interaction Analysis

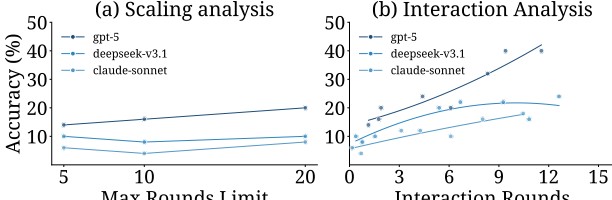

*Figure 4.* Model performance under different rounds constraints.

To test whether interaction underutilization stems from voluntary choice rather than capability deficits, we implement forced interaction protocols that require agents to ask a minimum number of clarifying questions (ranging from 2 to 10) before providing answers, as shown in Figure 4(b).

Results reveal dramatic model-specific differences. GPT-5 doubles its accuracy from 20% to 40% when compelled to ask 8 questions, confirming strong reasoning capabilities hindered by voluntary underuse of interaction. However, not all models benefit, Claude-Sonnet-4 shows modest gains. This demonstrates that effective information acquisition is a distinct capability varying significantly across architectures. The precise results are provided in Appendix C.1. We further analyze whether these additional questions are actually useful in Appendix B.5.

### 4.7. Error Attribution Analysis

We conduct error attribution on failed *Answer+Search+Interact* trajectories from three models (Claude-Sonnet-4, GPT-5, Gemini-2.5-Pro). We group failures into four categories: *identification errors* (17.1%–87.8%), where agents interact and search but still lock onto wrong candidates; *overconfidence errors* (1.0%–70.5%), where agents answer with minimal interaction; *ineffective questioning* (5.6%–8.8%), where questions target attributes absent from context or the wrong entity and yield "I don't know"; and *retrievability failure* (1.1%–5.6%), where evidence cannot be located despite multiple searches.

These results suggest two levels of interaction failure: *strategic* (deciding when to interact) and *tactical* (deciding what to ask). GPT-5 is dominated by strategic failures (70.5% overconfidence), while Claude-Sonnet-4 is dominated by tactical failures (87.8% identification errors); Gemini-2.5-Pro falls in between (23.4% overconfidence, 67.6% identification errors). Ablations with complete context reach 59–72% accuracy (§ 4.3), indicating that most failures stem from missing discriminative-attribute elicitation rather than reasoning limits. This also explains why forced interaction remains below the ceiling (71.5%): asking more is insufficient if the questions do not reveal the key distinguishing attributes. A case study is provided in Appendix C.2.

## 4.8. Longitudinal Study

Tracking 15 months of model development reveals a concerning divergence: while BrowseComp performance improved seven-fold (10% to 70%), INTERACTCOMP performance remained stagnant. Recent models like GPT-5, DeepSeek-R1, and GPT-4.1 cluster around 6-14% accuracy with minimal variation over time. This exposes a fundamental blind spot in AI development: rapid progress on search-focused tasks has not translated to progress in interaction-based problem solving. Without explicit focus on interaction capabilities, models advance in reasoning and retrieval while remaining primitive at recognizing ambiguity, a critical limitation for practical deployment. Figure 1 illustrates this stark contrast, showing BrowseComp's steep upward trajectory alongside INTERACTCOMP's flat performance across all evaluated models.

## 5. Conclusion

This paper presents INTERACTCOMP, a benchmark for evaluating a critical yet underexplored capability of search agents: recognizing and resolving ambiguous queries through interaction. Existing search benchmarks have driven major progress in retrieval and reasoning, but typically assume users provide fully queries from the outset, which diverges from real usage where information needs are often incomplete. INTERACTCOMP constructs instances that are easy to verify once the right context is obtained, yet fundamentally underdetermined without clarification, thereby testing whether agents can detect ambiguity and proactively ask for discriminative information during search.

Across 17 models, we find systematic overconfidence to be the dominant bottleneck. When given complete context, models reach 68–72% accuracy, but with interaction available they achieve only 13.73%, indicating that agents severely underuse clarifying questions despite having access. Natural-language interaction raises GPT-5 to 25.71%, yet a large gap to the with-context ceiling persists, suggesting that increased bandwidth alone does not solve the core difficulty of identifying what must be asked. Forced-interaction settings roughly double accuracy, confirming a strategic failure mode where latent capability is not engaged by current policies. Longitudinal results further expose this blind spot: while BrowseComp performance improved seven-fold over 15 months, INTERACTCOMP remained largely stagnant. Finally, because search provides grounded and verifiable outcomes, INTERACTCOMP offers clean reward signals and a practical substrate for reinforcement learning to train uncertainty-aware, actively interactive agents.Future work may further combine interaction training with reasoning-efficiency methods that reduce unnecessary long traces or redundant deliberation (Wu et al., 2025), enabling agents to ask more targeted questions while controlling computational cost.

## Impact Statement

This paper introduces InteractComp, a benchmark designed to advance the evaluation of multi-turn, user-centric agents under conditions of ambiguous information needs. By explicitly modeling interactive clarification and user feedback, InteractComp promotes the development of agents that can reason about uncertainty, ask targeted questions, and adapt their behavior based on partial or constrained user responses. These capabilities are particularly relevant for real-world applications such as information retrieval, decision support, and task-oriented assistants, where initial user intent is often underspecified. At the same time, enabling agents to engage in interactive questioning raises important considerations around user privacy, interaction fatigue, and potential over-optimization toward elicitation strategies. By providing a controlled and transparent evaluation setting, InteractComp encourages the study of interaction policies that balance information gain, user burden, and task accuracy, supporting more reliable and accountable human–AI collaboration.

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

# A. Benchmark and Implementation Details

## A.1. Instance Example

*Table A1.* A task instance from INTERACTCOMP. Tasks in INTERACTCOMP comprise an ambiguous query, the simulated user's context, and a concise answer.

| | |
|---|---|
| **Question:** Which team-based striking sport features two sides alternating offense and defense, where individuals sequentially hit a high-speed projectile and teammates coordinate to intercept it in the air? Outcomes depend on whether the projectile is intercepted or lands within the valid playing field. Defense relies on wide positioning and collaboration, all offensive players take turns striking, flight speeds often exceed 100 mph, protective gear is required due to impact risk, and the sport is governed by long-standing associations or leagues. | **Context:** Struck object is a plastic puck, resembling an ice hockey puck. Striking method uses a whip-like swing: the hitter lashes the puck with a long wooden rod. Defenders wield wooden boards, swinging them to block the puck in mid-air. Field is a giant fan shape, about 300 meters long with a 10–12 degree angle. Defensive teams deploy 18–20 players spread across the field to form a defensive line. Scoring is based on distance and landing point: offensive points depend on how far the puck travels and whether it touches the ground. |
| **Distractor:** *BaseBall* | **Answer:** *Hornussen* |

## A.2. Agent Implementation

### A.2.1. AGENT CONFIGURATIONS

Our agent implementation is built upon the ReAct framework (Yao et al., 2023), which combines reasoning and acting in a unified architecture. Our implementation records complete search, interaction, and answer trajectories for each rollout. Such trajectories may also serve as reusable supervision signals for future tool-use and agent-training methods, complementing recent work on synthesizing tool-use trajectories from textual experience (Xu et al., 2026). We implement four distinct agent configurations to systematically evaluate different capability combinations:

Configuration 1: Answer-only: The agent directly generates responses using its internal knowledge without external information gathering. This configuration serves as a baseline to measure pure knowledge recall capabilities on ambiguous queries.

Configuration 2: Answer+Search: The agent can perform web search actions to retrieve external information before generating answers. Available actions include:

- `search(query)`: Performs web search with the specified query
- `answer(response, confidence)`: Provides final answer with confidence score

Configuration 3: Answer+Interact: The agent that can additionally request clarification from users. This configuration adds:

- `interact(question)`: Poses yes/no questions to gather missing information

Configuration 4: Answer+Search+Ask: The full interaction-enabled agent that can additionally request clarification from users and get search results from web.

Table A2 summarizes all experimental configurations used in this work, including their available actions, response types, and round limits.

### A.2.2. SEARCH IMPLEMENTATION

We implement all search actions using the Google Serper API (Google Search wrapper). For each query, we send a POST request to the Serper endpoint (https://google.serper.dev/search) with a fixed timeout of 30 seconds. We restrict retrieval to the top 5 organic results exactly as configured in our agent code.

### A.2.3. ACTION SPACE DESIGN

Each agent operates with a maximum of 10 rounds, where each round allows exactly one action. The agent maintains an internal memory of previous actions and observations. For forced interaction experiments, we implement a constraint requiring minimum interaction thresholds before answer generation is permitted.

*Table A2.* Summary of all experimental configurations used in this work.

| Configuration | Search | Interact | Response Type | Round Limit | Location |
|---|:---:|:---:|:---:|:---:|:---|
| Answer-only | ✗ | ✗ | — | 1 | Table 2 |
| Answer+Search | ✓ | ✗ | — | 10 | Table 2 |
| Answer+Interact | ✗ | ✓ | Yes/No/IDK | 10 | Table 2 |
| With-context | ✓ | ✗ | — (given) | 1 | Table 2 |
| Answer+Search+Interact (**main**) | ✓ | ✓ | Yes/No/IDK | 10 | Table 1 |
| + Scaling variant | ✓ | ✓ | Yes/No/IDK | 5/10/20 | Table 3 |
| + Forced variant | ✓ | ✓ | Yes/No/IDK | 10, min $N$ | Figure 4 |
| + askNL variant | ✓ | ✓ | Free-form NL | 10 | Table 4 |

The complete system prompts and interaction protocols are detailed below.

## Prompt

```
SYSTEM_PROMPT = """
## Goal
You are an intelligent agent, designed to answer user's question.
In each round, you can execute one action, and you can get the action's result as observation.
You should think step by step, and output the action you want to execute.

### Evidence first
Before answering, you MUST:
1. Identify ALL missing information dimensions (time, scope, context, conditions etc.)
2. Systematically gather evidence for each dimension
3. Verify key assumptions through multiple sources/questions
4. Only answer when you can confidently justify each part of your response

**Critical**: Most questions have hidden complexities. Your initial understanding is likely incomplete.

### Using ask
When the ask action is available, you may pose closed-ended questions to fill gaps such as time, scope,
↪   conditions, relationships, or quantities.
- Do **not** ask the user to confirm a complete candidate answer or entity name. request neutral attributes
↪   or other missing evidence instead.

**Important: When you choose the ask action, you can only ask closed-ended, yes/no questions. The user will
↪   only respond with "yes", "no", or "I don't know".**

## Available actions:
{actions}

## Output Format
When you output the action,
you should output the action name and parameters in the json format, and only one action.
Such as,
```json
{{
    "action": "",
    "params": {{
        "<param_name>": "<param_value>"
    }}
}}
```
Before output, you should think step by step.

## Question
{question}
"""

ACT_PROMPT = """
## Memory
{memory}

## Observation
Last action: {last_action}
Observation: {last_observation}

## Question
{question}

## Action
```

```
You should output the action you want to execute.
Output your next action in JSON format, e.g.
```json
{{
    "action": "",
    "params": {{
        "<param_name>": "<param_value>"
    }}
}}
```

## ROUNDS
Current round: {round_info}
You have only one opportunity to provide your final answer.
Use your remaining rounds wisely to collect evidence and test your theories before committing to an answer.
The above shows your remaining action rounds.
"""

FINAL_ROUND_ACT_PROMPT = """

Given the question and information you have gathered, output the final answer.

## Round
{round_info}

## Memory
{memory}

## Question
{question}

## Action
You should output the answer action, you can think step by step before you output the answer.
Return the final answer action in JSON, for example:
```json
{{
    "action": "answer",
    "params": {{
        "answer": "<param_value>",
        "confidence": "<param_value>"
    }}
}}
```
"""
```

### A.3. Responder Simulation

We implement a controlled responder simulation using GPT-4o (temperature=1.0) that provides structured feedback when agents employ the *ask* action. Upon receiving agent queries, the responder evaluates questions against available context and responds with one of three standardized options: "yes", "no", or "I don't know". The responder state $s_r$ consists of the given context and interaction history, with transitions $T_r : (s_r, q_{agent}) \rightarrow o_r \in \{\text{yes, no, unknown}\}$ conditioned on context-question alignment. While maintaining response diversity through LLM generation, the constrained output format ensures evaluation consistency. INTERACTCOMP intentionally adopts this minimal interaction channel to prioritize diagnostic clarity over conversational realism.

The complete responder prompts are detailed below.

**Prompt**

```
RESPONDER_PROMPT = """
You are a specialized Q&A agent. Think step by step before you output the answer.

Rules:
- Reply with exactly one of: yes, no, or i don't know.
- Treat the context as the entire truth.
- Use only the provided CONTEXT to judge the yes/no question.
- Answer **yes** only if the context clearly states the proposition is correct.
- Answer **no** if the context contradicts the proposition (for example it states an incompatible
↪  attribute).
- If the context neither confirms nor denies it, answer **i don't know**.
- Do not rely on outside knowledge, analogies, or multi-hop guesses. Compare the relevant words directly.

CONTEXT
```

```
{context}

QUESTION
{question}

Output: yes | no | i don't know
"""
```

## A.4. Data Construction Pipeline

*Table A3.* Data Construction Pipeline: Step-by-Step Example

| Step | Component | Example Content |
|------|-----------|-----------------|
| Step 1 | Target Entity A | *Hornussen (Swiss team striking sport)* |
| | Distractor B | *Baseball (globally popular team bat-and-ball sport)* |
| Step 2 | Shared Attributes | Team-based striking game; offense/defense alternation; players take turns hitting; projectiles reach very high speeds ($>100$ mph); protective gear required; governed by formal associations or leagues. |
| | Distinctive Attributes | **Hornussen:** strikes a plastic puck ("Nouss") with whip-like swing using a long wooden rod; defenders intercept with wooden boards; fan-shaped field $\sim$300m; 18–20 defenders spread in wide formation; scoring depends on distance/landing point. |
| Step 3 | Ambiguous Question $Q$ | "Which team-based striking sport features two sides alternating offense and defense, where individuals sequentially hit a high-speed projectile and teammates coordinate to intercept it in the air? Outcomes depend on whether the projectile is intercepted or lands within the valid playing field. Defense relies on wide positioning and collaboration, all offensive players take turns striking, flight speeds often exceed 100 mph, protective gear is required due to impact risk, and the sport is governed by long-standing associations or leagues." |
| Step 4 | Contextual Information | – Struck object is a plastic puck, resembling an ice hockey puck.
– Striking method uses a whip-like swing with a long wooden rod.
– Defenders use wooden boards to block the puck in mid-air.
– Field: fan shape, $\sim$300m long, 10–12° angle.
– Defensive line: 18–20 players.
– Scoring: distance/landing-based. |
| Step 5 | Reasoning Path | $Q$ gives a plausible candidate set (e.g., Baseball vs Hornussen). Adding context clarifies unique Hornussen features (puck, whip swing, fan-shaped field, defensive boards), leading to the unique answer = Hornussen. |

## A.5. Evaluation Protocol

We validate simulation reliability through repeated sampling across identical context–question pairs across $k = 3$ trials, indicating stable behavior despite the stochastic process.

## A.6. Calibration Error (C.E.).

We evaluate confidence calibration using Expected Calibration Error (ECE). We divide the confidence interval $[0, 1]$ into $B = 5$ equal-width bins and, for each bin $b$, compute the empirical accuracy $\text{acc}_b$, the mean confidence $\text{conf}_b$, and their absolute gap. The final metric is:

$$\text{ECE} = \sum_{b=1}^{B} \frac{n_b}{N} \left| \text{acc}_b - \text{conf}_b \right|,$$

where $n_b$ is the number of samples in bin $b$ and $N$ is the total number of valid samples. Confidence values are normalized into $[0, 1]$ during parsing, with invalid values discarded.

# B. Additional Robustness Analyses

## B.1. Responder Sensitivity

We further test whether the choice of GPT-4o as the responder substantially affects the results. On a 50-question subset, we compare the default GPT-4o responder with a matched responder, where the answering model also serves as the responder.

*Table B1.* Accuracy with GPT-4o responder versus matched responder.

| Model | GPT-4o Responder | Matched Responder |
|---|---|---|
| GPT-5 | 14.00% | 20.00% |
| DeepSeek-R1 | 10.00% | 14.00% |
| Claude-Sonnet-4 | 6.00% | 10.00% |
| Gemini-2.5-Pro | 10.00% | 6.00% |
| GPT-4o-mini | 4.00% | 4.00% |

Although individual models vary, all models remain far below the with-context ceiling. This indicates that the main conclusion is not solely driven by the use of GPT-4o as the responder.

## B.2. Human Validation of the Simulated Responder

Because our evaluation relies on a simulated responder, we validate GPT-4o responses against human annotations. We sample 100 interaction records, each consisting of the hidden context, the agent's clarification question, and the simulator response. Two human annotators independently answer each clarification question with `yes`, `no`, or `I don't know` based only on the hidden context.

Agreement among GPT-4o and the two annotators reaches Krippendorff's $\alpha = 0.707$. Cohen's $\kappa$ between GPT-4o and annotator 1 is 0.756, and Cohen's $\kappa$ between GPT-4o and annotator 2 is 0.693. These results indicate acceptable agreement for a controlled evaluation setting, while also suggesting that simulated feedback should be viewed as an approximation rather than a replacement for real user interaction.

## B.3. Commercial DeepResearch Systems

To test whether the observed failure is merely an artifact of our ReAct scaffold, we additionally evaluate two commercial search-agent systems with richer agentic workflows. Doubao Deep Search achieves 10.0% accuracy and OpenAI Deep Research achieves 6.67% on the full benchmark. Both remain within the same low-accuracy band as our ReAct-based agents, suggesting that richer agentic scaffolding alone does not eliminate the ambiguity-resolution gap.

## B.4. Ranking Consistency Across Runs

To assess whether model ordering is stable across independent trials, we compute Kendall's $\tau$ between the three main runs.

The rankings show statistically significant consistency across runs. However, we emphasize that our main conclusion does not depend on exact point-wise rankings, but on the large gap between full interaction performance and the with-context ceiling.

*Table B2.* Ranking consistency across independent runs.

| Comparison | Kendall's $\tau$ | $p$-value |
|---|---|---|
| Trial 1 vs. Trial 2 | 0.428 | 0.020 |
| Trial 1 vs. Trial 3 | 0.595 | 0.001 |
| Trial 2 vs. Trial 3 | 0.557 | 0.002 |

## B.5. Interaction Utility Analysis

Interaction frequency alone does not indicate whether clarification is useful. We therefore compute three utility-oriented metrics on a 50-question subset: average belief change per ask, post-ask correctness gain, and usefulness rate. The results show that more questions do not necessarily lead to better disambiguation. For example, GPT-4o-mini asks 2.4× more questions than GPT-5, yet achieves only about one-fifth of GPT-5's post-ask correctness gain. This explains why high interaction rate does not translate into high accuracy in Table 1 and suggests that future training should reward useful clarification rather than raw question frequency.

*Table B3.* Interaction utility metrics on a 50-question subset.

| Model | Ask Count | $\Delta$Belief/Ask | Correctness Gain | Usefulness Rate |
|---|---|---|---|---|
| GPT-5 | 113 | 6.53 | 11.80% | 38.10% |
| DeepSeek-R1 | 153 | 7.41 | 3.30% | 17.00% |
| GPT-4o-mini | 269 | 5.02 | 2.60% | 21.50% |
| Gemini-2.5-Pro | 30 | 3.00 | 8.00% | 23.30% |
| Claude-Sonnet-4 | 53 | 1.60 | 2.00% | 35.80% |

# C. Additional Experimental Results and Case Studies

## C.1. Rounds Constraints Results

*Table C1.* Performance comparison of models under varying average interaction levels. Metrics include accuracy (%), expected calibration error (ECE, %), and average rounds of interaction (Interaction).

| Interaction | Accuracy (%) | ECE (%) | Setting |
|---|---|---|---|
| *GPT-5* | | | |
| 1.14 | 14.0 | 71.50 | SCALING |
| 1.76 | 16.0 | 71.54 | SCALING |
| 1.90 | 20.0 | 70.06 | SCALING |
| 4.40 | 24.0 | 63.34 | FORCED |
| 6.10 | 20.0 | 69.02 | FORCED |
| 8.32 | 32.0 | 54.68 | FORCED |
| 9.40 | 40.0 | 48.20 | FORCED |
| 11.56 | 40.0 | 46.86 | FORCED |
| *DeepSeek-Chat* | | | |
| 0.38 | 10.0 | 77.00 | SCALING |
| 0.74 | 8.0 | 80.30 | SCALING |
| 1.54 | 10.0 | 75.20 | SCALING |
| 5.40 | 20.0 | 62.30 | FORCED |
| 6.68 | 22.0 | 52.60 | FORCED |
| 9.26 | 22.0 | 61.30 | FORCED |
| 10.82 | 16.0 | 66.40 | FORCED |
| 12.62 | 24.0 | 61.20 | FORCED |
| *Claude-Sonnet-4* | | | |
| 0.16 | 6.0 | 79.90 | SCALING |
| 0.70 | 4.0 | 80.24 | SCALING |
| 0.78 | 8.0 | 81.84 | SCALING |
| 3.12 | 12.0 | 75.80 | FORCED |
| 4.26 | 12.0 | 76.90 | FORCED |
| 6.10 | 10.0 | 76.00 | FORCED |
| 8.02 | 16.0 | 69.10 | FORCED |
| 10.46 | 18.0 | 68.40 | FORCED |
| *GPT-4o-mini* | | | |
| 2.00 | 4.0 | 49.50 | SCALING |
| 3.62 | 8.0 | 47.60 | SCALING |
| 2.76 | 8.0 | 33.20 | SCALING |
| 14.50 | 4.0 | 65.50 | FORCED |
| 16.50 | 4.0 | 69.70 | FORCED |
| 14.88 | 2.0 | 62.60 | FORCED |
| 11.18 | 6.0 | 56.10 | FORCED |
| 14.92 | 2.0 | 66.70 | FORCED |

## C.2. Case Study On Failed Case.

To illustrate the mechanism of errors (failures in § 4.7), we examine a representative case from the Sports domain where the agent performed both interaction and search but failed to identify the correct answer. The agent demonstrated willingness to gather information, asking four clarification questions and performing three searches. However, all actions revolved around hockey-like attributes (puck, sticks, ice surface) rather than discriminative features unique to Hornussen.

*Table C2.* Case study from the **Sports** domain showing how insufficient clarification leads to failure. The correct answer is *Hornussen*, but the model incorrectly concludes that it is Ice Hockey.

| Field | Content |
|---|---|
| Question | Team-based striking sport where players sequentially hit a high-speed projectile (>100 mph) and defenders attempt mid-air interception. Scoring depends on whether the projectile is intercepted or lands in the valid field. Requires protective gear and is governed by long-standing associations. |
| Answer | *Hornussen* |
| Model prediction | "Ice Hockey." (Confidence: 75) |
| Actions | **Ask 1:** "Is the projectile a ball?" → **No**
**Ask 2:** "Is the projectile a puck?" → **Yes**
**Ask 3:** "Do players use sticks to hit the puck?" → **Yes**
**Ask 4:** "Is this sport played on ice?" → **I don't know**
**Search 1:** "puck sport alternating offense defense sequential hitting air interception"
**Search 2:** "puck sport 100 mph protective gear turns"
**Search 3:** "hockey variant alternating turns interception"
(All retrieved results were hockey-related.) |

