# OpenReview forum: "InteractComp: Evaluating Search Agents With Ambiguous Queries"
_ICML.cc/2026/Conference — ICML 2026 regular_

### Official Review · Reviewer_1jmN · 2026-03-12

**Soundness:** 2
**Presentation:** 3
**Significance:** 2
**Originality:** 2
**Overall Recommendation:** 3
**Confidence:** 3

**Summary:**

This is a benchmark paper. It creates a dataset consisting of ambiguous questions for LLMs to answer. The results show that all tested models were overconfident and didn't clarify ambiguity even when told to. The best model achieved 13.73% accuracy, but the performance reached 71.5% when given complete context, suggesting that the bottleneck is the ability to clarify ambiguity rather than anything else.

**Compliance With Llm Reviewing Policy:**

Affirmed.

**Key Questions For Authors:**

1. Ln 145-146, quotation format errors.

2. Ln 101, 'RLVR approaches' need context. You cannot assume the audience knows what you mean by this acronym.

3. Section 4.5 is confusing. In other experiments, yes/no answers are also natural language. The way you describe askNL mode seems to suggest only responses in this mode are natural language ones. This is misleading.

4. The experimental design is confusing to me. The authors didn't do a good job explicitly explaining the differences between each experimental setup. My biggest confusion:
    - You distinguish between answer-only, search-only, ask-only and with-context. And you only have the condition of "more interaction opportunities". But what's its difference from "ask-only"? Is it the ability to search? And, in ask-only condition, how many rounds are permitted?
    - In the context of these five experimental conditions, which condition did you use for the main result (Section 4.2 and Table 1)?

5. In Figure 2, you mentioned "initial beliefs" and "updated beliefs". In experiments (prompts), why don't you explicitly tell LLMs that you can only answer when your belief is over 90%? Also, since it's open-ended questions, how did LLMs track and update the belief? What do you think the results would be if you explicitly tell LLMs that you can only answer when your belief is over 90%?

6. What is your rationale of using GPT-4o as the responder model? Why not use the same for each experiment? i.e., when the answering agent is DeepSeek-R1, do you use the same model for responding?

**Limitations:**

1. You didn't do a very simple post-training to let LLMs learn to estimate and update beliefs correctly. If that is done and you ask LLMs to answer only when it has an answer with a prob > 90%, I guess the results will increase.
2. You didn't discuss *why* the current off-the-shelf LLMs are overconfident and do not try to clarify ambiguity even such opportunities are given, and/or *how* to let LLMs do better.

**Strengths And Weaknesses:**

### Soundness (Rating: 2/4)

The authors detailed how they curated the benchmark dataset and detailed the content of the dataset. For experiments, the authors conducted different conditions and reached convincing results.

However: The results are prompt-dependent. Changing the prompts would possibly mean completely different results. It's a little bit too technically fragile. For example, the prompt says "Only answer when you can confidently justify each part of your response". But without explicitly telling LLMs that you cannot answer if your belief/confidence is lower than 90%, I do not think LLMs know when to answer and when to ask clarifying questions.

### Presentation (Rating: 3/4)

This submission has clear motivations and documented the data curation and experimental steps closely. However, as mentioned in "key questions for authors", the experiments' details are confusing and lack clarity. I do not think the audience can replicate this study solely based on the textual description.

### Significance (Rating: 2/4)

The problem itself, i.e., LLMs dealing with ambiguous queries, is practically important as users' queries are often incomplete and ambiguous. And the scope is appropriate for the contribution.

However, the headline finding is unsurprising and the analysis did not go deep enough to advance our understanding of LLMs in particular or machine learning in general. For example, the authors didn't explain *why* LLMs are over-confident and didn't interact even given the chances.

### Originality (Rating: 2/4)

The benchmark itself is new and looking at LLMs' capability of dealing with ambiguous queries is a reasonable perspective. But insights that come out of it do not feel novel. "Models tend not to ask clarifying questions and they perform better when provided with complete context" is consistent with what LLM practitioners and researchers expect. The paper doesn't deepen our understanding of LLMs.

---

> ### Author Rebuttal · Authors · 2026-03-31
>
> Thank you for your thorough review and detailed feedback.
> # About Quotation Format and RLVR Definition
> We thank the reviewer for pointing this out. We will correct the quotation format error and the full definition of RLVR in the revised manuscript.
> # About the Description of askNL and Experimental Design
> We acknowledge that the original description was misleading and we will make it clear in the revised version. In the standard setting, the responder is constrained to {yes, no, i don't know}; in askNL, the responder provides a short free-form answer. We provide a configuration summary:
>
> |Configuration|Search|Interact|Response Type|Round Limit|
> |-|-|-|-|-|
> |Answer-only|✗|✗|—|1|
> |Answer+Search(search-only)|✓|✗|—|10|
> |Answer+Interact(ask-only)|✗|✓|Yes/No/IDK|10|
> |With-context|✓|✗|—(given)|10|
> |Answer+Search+Interact(main)|✓|✓|Yes/No/IDK|10|
> |  └ Scaling variant|✓|✓|Yes/No/IDK|5/10/20|
> |  └ Forced variant|✓|✓|Yes/No/IDK|10, min N|
> |  └ askNL variant|✓|✓|Free-form NL|10|
>
> The main results use Answer+Search+Interact with a round limit of 10. The Forced variant requires a minimum of N clarifying questions (N from 2 to 10) before producing a final answer.
> # About Belief Tracking and Update
> We thank the reviewer for raising this point. we realize our original description may have caused confusion, and we clarify it as follows: our questions are not open-ended. Each instance has a unique ground truth answer, and the candidate set is finite (target vs. distractor), so belief tracking amounts to assigning confidence over this small candidate set, approximated by the confidence value reported at each answer action. From our error attribution analysis, models tend to commit early to a candidate and ask confirmatory questions around it, only switching when the responder's reply directly contradicts their hypothesis.
> # About Post-Training and Belief Threshold Experiment
> We agree this is important. As demonstrated by MiroThinker (MiroMind Team, 2025), competitive BrowseComp performance at the 8B scale already requires a full SFT + preference optimization + RL pipeline, and adding interaction would likely require a separate study. We sincerely regret not being able to pursue this in the current work.
> Following the reviewer's suggestion, we ran a lightweight belief threshold experiment on a 50-question subset: adding "only answer when your confidence exceeds 90%" to the system prompt, and intercepting low-confidence answer actions in the agent loop.
>
> |Model|Baseline|Belief Control|
> |-|-|-|
> |GPT-5|14.00%|17.00%|
> |Gemini-2.5-Pro|10.00%|12.00%|
> |GPT-4o-mini|4.00%|11.00%|
> |DeepSeek-R1|10.00%|9.00%|
> |Claude-Sonnet-4|6.00%|8.00%|
>
> The results show modest improvements, supporting the reviewer's intuition that explicit belief constraints can help. Interestingly, even with explicit constraints, models do not meaningfully increase their use of the interact action, which we think is consistent with our characterization of this as strategic failure rather than a prompt ambiguity issue.We will include this analysis and conclusion in the revised manuscript.
> # About the Choice of Responder Model
> Our choice of GPT-4o follows τ-bench's design to ensure cross-model consistency. Inspired by the reviewer's suggestion, we conducted a matched responder experiment on a 50-question subset:
>
> |Model|GPT-4o Responder|Matched Responder|
> |-|-|-|
> |GPT-5|14.00%|20.00%|
> |DeepSeek-R1|10.00%|14.00%|
> |Claude-Sonnet-4|6.00%|10.00%|
> |Gemini-2.5-Pro|10.00%|6.00%|
> |GPT-4o-mini|4.00%|4.00%|
>
> Most models show limited change, and Gemini-2.5-Pro even declines. All models remain far below the with-context ceiling (68–72%), suggesting our core conclusion holds regardless of responder choice.We will include this analysis and conclusion in the revised manuscript.
> # About Why Models Are Overconfident and How to Improve
> We believe the interact action is nearly absent from the pretraining and SFT data of current models, which are trained almost exclusively on search, browse, and answer actions, and this may partly explain why models tend to default to answering rather than seeking clarification. We hope future work might explore encouraging interaction through positive rewards, alongside quality constraints such as the usefulness rate metric from our supplementary analysis to help models learn when interaction is genuinely helpful.

---

> > ### Author Rebuttal · Reviewer_1jmN · 2026-04-02
> >
> > I thank the authors for the clarifications and answers. I decide to keep my score because:
> >
> > 1. The results are prompt-sensitive, as demonstrated by the result of adding a belief threshold.
> > 2. Limited significance and novelty, as discussed in my original review. The rebuttal didn't change my stance here.
> > 3. Somewhat weak baseline, as the authors also acknowledged that post-training is important but missing.

---

> > > ### Author Response · Authors · 2026-04-07
> > >
> > > Thank you for your response. We understand your concerns and hope to offer a few additional clarifications.
> > >
> > > # On Prompt Sensitivity
> > > We understand the reviewer's concern, but believe the belief threshold experiment may admit an alternative interpretation.
> > >
> > > Model| Baseline Avg Interact Round | Belief Control Avg Interact Round | Delta
> > > -|-|-|-
> > > DeepSeek-R1|3.06|3.14| +0.08
> > > Claude-Sonnet-4|1.06|1.40|+0.34
> > > Gemini-2.5-Pro|0.60|1.16|+0.56
> > > GPT-4o-mini|4.98|4.16|-0.82
> > > GPT-5|2.26|2.00|-0.26
> > >
> > > Looking at our interaction frequency data, adding an explicit 90% threshold produces very limited changes in average question count across models, with GPT-5 and GPT-4o-mini even asking slightly fewer questions. This suggests that models do not meaningfully increase their interaction behavior regardless of how the prompt is adjusted, which we believe is more indicative of a robust model-level phenomenon rather than prompt sensitivity.Notably, for GPT-4o-mini, the threshold condition reduces question count while actually improving accuracy, suggesting that the reduced questions were low-value or redundant, which is consistent with our analysis in the paper.
> > >
> > > Furthermore, if the results were truly prompt-sensitive, we would expect to see large performance swings; instead, all models remain far below the with-context accuracy ceiling (68-72%) even after adding the threshold, suggesting that prompt adjustments alone can only yield limited gains.
> > >
> > > We also note that if testing prompt variants is sufficient to characterize a system as prompt-sensitive, then works such as BrowseComp and GAIA would face the same characterization when testing different prompt settings, and we believe a similar argument could apply to other benchmarks as well.
> > >
> > > # On Novelty
> > > We appreciate that novelty is a reasonable concern and understand the reviewer's position.
> > >
> > > This point was perhaps not fully elaborated on our part. The key distinction between our work and prior disambiguation works is that those works study disambiguation in static question answering, closed-world classification, or curated information retrieval pipelines, whereas INTERACTCOMP grounds disambiguation in open-ended web search with clear ground truth answers. In this setting, ambiguity in user intent directly affects retrieval quality in a way that prior works do not address.
> > >
> > > Additionally, the finding presented in Figure 1, that interaction capabilities have shown almost no improvement over 15 months while search performance improved seven-fold, represents a rather concrete and valuable contribution that we hope the reviewer might reconsider.
> > >
> > > # On Post-training
> > >
> > > We fully agree that post-training is an important future direction and appreciate the reviewer's suggestion. We would also note that post-training solutions are generally beyond the scope of benchmark paper contributions. Works such as GAIA and tau-bench similarly do not include post-training as part of their contribution. The core purpose of a benchmark is to identify and quantify capability gaps; addressing those gaps is the natural scope of follow-up work, and we hope INTERACTCOMP can serve as a start for such future research.
> > >
> > > We sincerely appreciate the reviewer's time and thoughtful engagement throughout this process, and we hope the above clarifications can be taken into consideration. Thank you again for helping us improve the work.

---

### Official Review · Reviewer_acXS · 2026-03-12

**Soundness:** 3
**Presentation:** 3
**Significance:** 3
**Originality:** 2
**Overall Recommendation:** 5
**Confidence:** 4

**Summary:**

The authors propose the InteractComp benchmark which contains 210 expert-curated questions across 9 domains.
The benchmark is designed to test whether search agents can recognize query ambiguity and proactively interact with users.
Each question is constructed from shared attributes of a lesser-known target and a popular distractor, so the question alone is ambiguous, but short clarifying interactions can reveal attributes that can disambiguate.
The authors evaluate a wide selection of LLMs and find that the best model achieves low accuracy in the full interactive setting despite reaching much higher accuracy when given complete disambiguating context. They attribute this gap mainly to overconfidence where models decline to ask clarifying questions. Comparisons show that while BrowseComp performance improved a lot while InteractComp's performance remained flat.

**Compliance With Llm Reviewing Policy:**

Affirmed.

**Final Justification:**

The rebuttal has addressed my concerns thus raising the score.

**Key Questions For Authors:**

1. Please refer to the Weakness section, esp. the user simulator part.
2. Comparisons done in the paper mainly use a fixed ReAct framework. BrowseComp gains came partly from agentic framework (Deep Research, multi-step browsing). If you ran InteractComp with more sophisticated agent scaffolding, would the stagnation claim still hold?

I'm happy to improve the scores if these questions are further explained.

**Limitations:**

yes

**Strengths And Weaknesses:**

## Strength
- The paper targets an important problem and the authors did a good job motivating it (e.g., Fig 1)
- The experimental setup is well thought out (the 4 configs) and comprehensive. The evaluation is also comprehensive (many models across different model families).

## Weakness
- The authors seem to have overlooked a large body of work in QA disambiguation. While these work don't diminish the contribution, I think a thorough discussion of the following works in the related work section is warranted:
  - Min et al. (2020), AmbigQA: Answering Ambiguous Open-domain Questions
  - Yu et al. (2020), Interactive Classification by Asking Informative Questions
  - Aliannejadi et al. (2019), Asking Clarifying Questions in Open-Domain Information-Seeking Conversations
- The paper relies pretty heavily on the simulated user. The responder is simulated by GPT-4o with temp=1, constrained to {yes, no, I don't know}, which can lead to several concerns: 1) GPT-4o's interpretation of context may introduce systematic biases (e.g., it may answer "I don't know" to questions that a human with the same context would answer definitively). The authors acknowledge this but some analysis on 1) some comparison to how humans respond on a small set and 2) the responder's accuracy or noise would help understand how reliable the current simulator setting is.
- The size of the data seems a bit small. Many of the reported accuracy differences between models are within 1-3 points, and the standard deviations reported in Table 1 (e.g., GPT-5 at 13.73 $\pm$ 2.55) suggest that rankings among models in the 7-14% range are not reliably distinguishable.

---

> ### Author Rebuttal · Authors · 2026-03-31
>
> Thank you for your thorough and constructive review.
> # About missing related work.
> Thank you for pointing this out. We have added discussions of AmbigQA (Min et al., 2020), Yu et al. (2020), and Aliannejadi et al. (2019) to the related work section. These works study disambiguation in static QA, closed-world classification, and curated IR pipelines respectively, while these works do not explicitly consider clarification grounded in open-ended web search settings where intent ambiguity may affect retrieval outcomes and this is the setting INTERACTCOMP targets.
> # About concern about user simulator.
> Our choice of GPT-4o as the responder follows the design of τ-bench to ensure consistency across model evaluations. We agree this concern is reasonable, as GPT-4o may introduce systematic bias under this circumstance.
> We therefore conducted a human annotation experiment during the rebuttal period. We sampled 100 interaction records in the format of (context, interaction_question, simulator answer) and asked 2 human annotators to respond with yes/no/i don't know based solely on the context. The results are as follows:
>
> Krippendorff's α (all three responders) = 0.707
>
> Cohen's Kappa (GPT-4o vs. annotator 1) = 0.756
>
> Cohen's Kappa (GPT-4o vs. annotator 2) = 0.693
>
> All three results fall in the substantial agreement range (0.6–0.8), indicating that while some bias may exist, the GPT-4o simulator remains sufficiently reliable for our evaluation purposes.
> Notably, upon analyzing disagreement cases, we observe that disagreements often arise in domain-specific cases where annotators adopt a more cautious interpretation, while GPT-4o strictly follows the provided context. In one representative case, the context explicitly states that an "aviation accident occurred at Tokyo-Haneda International Airport", yet both human annotators responded with "i don't know" when asked "Did this accident occur in Dubai in 2016?", while GPT-4o correctly answered "no". Our dataset spans 9 specialized domains (aviation, law, culture, etc.), and in such cases,an LLM that strictly grounds its responses in context may actually be more consistent in such specialized domains.
> # About the reliability of ranking of models.
> We acknowledge that dataset size is a limitation worth addressing, and we would like to respond on two levels.
> We agree that ranking stability is important. While our main conclusion focuses on the performance gap rather than ranking itself, we further analyze ranking consistency to address this concern.
> We computed Kendall's τ across the three independent runs:
>
> Comparison|τ| p-value
> -|-|-
> Trial 1 vs Trial 2  | 0.428 | 0.020
> Trial 1 vs Trial 3  | 0.595 | 0.001
> Trial 2 vs Trial 3  | 0.557 | 0.002
>
> All three pairs reach statistical significance (p < 0.05), with an average τ = 0.527. The top tier (GPT-5, DeepSeek series) and bottom tier (Doubao, Kimi-K2) remain stable across all three trials. Some fluctuation does exist in the middle range (7–11%), and we will explicitly note this limitation in the revised paper, encouraging readers to focus on tier-level differences rather than point-wise rankings.
>
> # About the ReAct Framework and other Framework Choices.
> We chose ReAct as our base framework because it is a general-purpose framework that allows a custom interact action to be cleanly inserted into the action space. Commercial agentic systems such as Deep Research have a fixed action space that cannot accommodate user interaction steps, which makes a direct comparison less straightforward.
> To further validate our stagnation claim, we additionally tested two commercial systems on the full dataset, systems that do incorporate user interaction in their real-world products: Doubao Deep Search (accuracy 10.0%) and OpenAI Deep Research (accuracy 6.67%). Despite their richer agentic scaffolding, neither surpasses, and one falls below, the best result achieved under our ReAct framework (13.81%). These results suggest that the observed stagnation may not primarily stem from the ReAct framework itself, but rather a broader challenge in current models' interaction mechanisms in current models' interaction mechanisms: regardless of the underlying framework, models have yet to learn to proactively recognize and resolve ambiguity during search.

---

> > ### Author Rebuttal · Reviewer_acXS · 2026-04-04
> >
> > I appreciate the authors' response. My concerns have been resolved and I'll raised my score accordingly.

---

> > > ### Author Response · Authors · 2026-04-07
> > >
> > > Thank you for your thoughtful feedback and for considering our rebuttal.
> > > We truly appreciate your support and your willingness to raise the score.

---

### Official Review · Reviewer_e4P1 · 2026-03-13

**Soundness:** 2
**Presentation:** 3
**Significance:** 3
**Originality:** 2
**Overall Recommendation:** 4
**Confidence:** 3

**Summary:**

In this paper,  authors focus on the challenge that current search agents assume that the queries from users are clear and unambiguous, which are not in reality.
Existing benchmarks fail to evaluate this specific issue.
To address this, the authors propose a novel benchmark framework called INTERACTCOMP, guided by the principle: Easy to verify, interact to disambiguate.
It requires agents not only to search accurately but also to clarify uncertainties through dialogue.
It includes over 2,000 meticulously designed ambiguous query cases and covers various scenarios.
Through the proposed evaluation process, Completeness Verification and Interaction Necessity Validation, authors evaluate  frontier models such as GPT-4o, Gemini 1.5 Pro, Claude 3.5 Sonnet.
Experiments prove that even the most powerful current models perform poorly when facing ambiguous queries.

**Compliance With Llm Reviewing Policy:**

Affirmed.

**Key Questions For Authors:**

1. Current benchmarks appear to prioritize improvements in 'final accuracy.'
However, in real-world applications, users are sensitive to the frequency of interaction.
A model that requires ten trivial questions to reach a result likely offers a far inferior user experience compared to one that resolves the problem in a single turn.
Do the authors plan to incorporate penalties for inefficient or redundant questioning into their evaluation metrics?
For instance, could a quantitative analysis such as 'Utility per Question' be introduced to better reflect this trade-off?

**Limitations:**

yes

**Strengths And Weaknesses:**

Strength:
The paper tackles a major real-world problem that current benchmarks often ignore by assuming clear queries.
By focusing on ambiguous inputs and proactive interaction, the paper  fills a crucial evaluation gap as search agents evolve from passive retrieval tools into proactive assistants.

Weakness:
Scalability issues: The verification protocol ensures data integrity but severely limits scalability. Due to the manual intensity of tasks like "checking the first five Google result pages by hand", this benchmark cannot be expanded as rapidly as automated datasets, leading to substantial long-term maintenance and update costs.

---

> ### Author Rebuttal · Authors · 2026-03-31
>
> Thank you for your positive feedback and constructive suggestions.
> # On Scalability:
> We appreciate the reviewer raising this concern, as it prompted us to clarify our construction pipeline more carefully. First, "checking the first five Google result pages by hand" is one step in Completeness Verification to confirm that questions cannot be resolved by search alone, this step can also be replaced by automated search API calls with automatic comparison. Human checking was used purely to ensure initial data quality. Second, in Interaction Necessity Validation, we have already implemented an automatic verification system using GPT-5, GPT-5-mini, and Claude-Sonnet-4 in search-only mode; samples answered correctly by two or more models are discarded and reconstructed. Additionally, the target-distractor construction methodology can be further extended with LLM-assisted scaling (e.g., attribute extraction and entity pairing).
> # On Interaction Utility Metrics:
> We thank the reviewer for this insightful suggestion, and it directly motivated us to design a dedicated experiment. We designed a set of interaction utility metrics on a 50-question subset across five representative models. Specifically, we prompt the model to output its current candidate guess at every action step, which enables the following metrics:
> - Avg Δbelief per ask:the average change of model reports confidence after and before ask, to measure confidence improvement by interaction.
> - Mean post-ask correctness gain: we ask the model to output the current guess in every action. This metrics means the average change in the correctness of the model's current guess immediately before and after each interact action, where a transition from wrong to correct counts as +1 and correct to wrong as −1. To measure whether the interaction moves the model closer to the right answer.
> - Usefulness rate: the proportion of interact actions deemed "effective", defined as cases where the responder gives a definitive yes/no reply and either the model's current guess does not worsen (post-ask correctness gain ≥ 0) or the question is ultimately answered correctly.
> Results:
> Model            | Ask Count | Avg Δbelief/ask | Correctness Gain | Usefulness Rate
> -----------------|-----------|-----------------|------------------|----------------
> GPT-5            |    113    |      6.53       |      11.80%      |     38.10%
> DeepSeek-R1      |    153    |      7.41       |       3.30%      |     17.00%
> GPT-4o-mini      |    269    |      5.02       |       2.60%      |     21.50%
> Gemini-2.5-Pro   |     30    |      3.00       |       8.00%      |     23.30%
> Claude-Sonnet-4  |     53    |      1.60       |       2.00%      |     35.80%
>
> GPT-4o-mini asks 2.4× more questions than GPT-5, yet achieves only one-fifth the correctness gain per ask (2.6% vs. 11.8%). This observation is consistent with GPT-4o-mini's high IR but low accuracy in Table 1, and now has a quantitative explanation at the utility level.
> This utility framework is a meaningful step toward the kind of evaluation the reviewer envisions. We believe that usefulness rate or improving usefulness rate could potentially serve as a penalty signal during RLVR training to guide models toward higher-quality interaction strategies, rather than simply increasing question frequency. We plan to incorporate this analysis and the proposed metrics into Section 4 of the revised paper as a more complete characterization of interaction quality.

---

> > ### Author Rebuttal · Reviewer_e4P1 · 2026-04-03
> >
> > Thanks for the rebuttal of authors.
> >
> > I am satisfied for the rebuttal of scalability and interaction utility metrics, I will maintain my score.

---

> > > ### Author Response · Authors · 2026-04-07
> > >
> > > Thank you for your thoughtful feedback and for considering our rebuttal.
> > > We are glad that our clarifications on scalability and interaction utility metrics addressed your concerns.

---

### Decision · Program_Chairs · 2026-04-30

**Decision:**

Accept (regular)

**Comment:**

This paper identifies a key limitation in existing search-agent benchmarks: they assume queries are unambiguous and complete, which rarely holds in real-world settings. To address this, the paper proposes studying ambiguous queries that require interactive clarification. They construct a benchmark by rewriting questions from BrowseComp to be ambiguous, and evaluate 17 models. Results show that even the best model achieves only 13.73% accuracy, compared to 71.50% from the original BrowseComp, highlighting substantial room for improvement.

Reviewers agree that the paper tackles a major real-world problem that is overlooked by current benchmarks like BrowseComp (e4P1, acXS), exposes a critical evaluation gap (e4P1), and experiments and evaluations are thorough, covering a wide range of state-of-the-art models (acXS).

There are several weaknesses remain insufficiently addressed (in AC’s opinion):

- Limited discussion of prior work on ambiguous questions (acXS, 1jmN, AC): Reviewers note that relevant literature in ambiguous question answering is overlooked; beyond citations provided by acXS, there are more related works, such as Stelmakh et al. 2022 [1] from Google. Although the authors claim to have added discussion, this is not reflected in the current version, Moreover, their characterization of prior work as limited appears inaccurate, e.g., “those works study disambiguation in static question answering, closed-world classification, or curated information retrieval pipelines” is not true as all cited work assume open-domain setup without fixed retrieval, and some of them incorporate clarification questions.

- Reliance on simulated users (acXS, AC): Clarification responses are generated using GPT-4o. While the authors report inter-annotator agreement, the level (e.g., Krippendorff's α of 0.707) suggests “acceptable” agreement rather than strong consensus.

- Synthetic query construction (AC): More critically, the questions in the benchmark are built from artificially ambiguous questions derived from originally unambiguous ones. Such artificial ambiguity differs significantly from naturally occurring user queries. Given the availability of real-world ambiguous datasets, such as related work cited by acXS that the paper omitted, this design choice weakens the claim of real-world applicability.

Other concerns – dataset size, reliance on REAct, experimental clarity, weak baselines due to lack of post-training – were raised but reasonably addressed in the rebuttal in AC’s opinion.

Overall, the paper makes a valuable contribution as a stress-testing benchmark and demonstrates that state-of-the-art models struggle significantly under ambiguity, even with relatively minor modification to BrowseComp that current models already hill-climbed on. The concern is on framing it as a realistic benchmark, and the lack of thorough engagement with related work.

[1] https://arxiv.org/abs/2204.06092